# Citizen Science for Water Quality Monitoring in the Meki River, Ethiopia: Quality Assurance and Comparison with Conventional Methods

**Wudinesh Zawuga Babiso [1,\*], Kinfe Kassa Ayano [1], Alemseged Tamiru Haile [2], Demissie Dawana Keche [3], Kishor Acharya [4] and David Werner [4]**

[1] Department of Water Supply and Environmental Engineering, Arba Minch Water Technology Institute (AWTI), Arba Minch University, Arba Minch P.O. Box 21, Ethiopia

[2] International Water Management Institute (IWMI), Addis Ababa, Ethiopia

[3] Water Resources Research Center, Arba Minch Water Technology Institute (AWTI), Arba Minch University, Arba Minch P.O. Box 21, Ethiopia

[4] School of Engineering, Newcastle University, Newcastle upon Tyne NE1 7RU, UK

\* Correspondence: 21wzawuga@gmail.com; Tel.: +251-955945599

**Abstract:** A lack of water quality information for many water bodies around the world makes it difficult to identify global change and discover early signs of myriad threats to freshwater resources. This problem is widely seen in Ethiopia due to absence of regular monitoring. Citizen science has a great potential to fill these gaps in water quality data, but there is concern about the accuracy of data collected by citizen scientists. Moreover, there is a gap to engage citizen scientists in water quality monitoring, and there is still insufficient awareness of how citizen scientists can become part of a collaborative scheme. This study aimed to evaluate the accuracy of water quality collected by citizen scientists and characterize the water quality of the Meki River with the involvement of citizen scientists. The suitability of the river water for irrigation was evaluated using a combination of citizen science and conventional water quality data collection methods. Water temperature, turbidity, ammonia, phosphate, nitrate, nitrite, total alkalinity, total hardness, and pH were analyzed by both citizen scientists and in a conventional laboratory. The citizen scientists' data, expressed as percent of synthetic standard solution concentrations, indicated good agreement for selected water quality parameters: $123.8 \pm 24.7\%$ for $PO_4^{3-}$, $115.6 \pm 6.3\%$ for $NO_3^-$, $105.8 \pm 7.4\%$ for pH, and $133.3 \pm 23.6\%$ for $NH_4^+$. Thus, citizen scientists can monitor and collect water quality data accurately. From the results, the Meki River water can be used for irrigation, but pollution sources should be controlled to reduce further quality deterioration as the population increases.

**Keywords:** citizen science; validation; irrigation; LaMotte color strips; water quality

## 1. Introduction

Water is essential for life, all aspects of socio-economic development, and healthy ecosystems. While there are sufficient freshwater resources at the global level to enable agricultural and industrial development, the long-term sustainable use of water resources is a growing concern [1]. One of the main threats to water resources is water pollution. However, monitoring water quality with conventional methods is expensive and time-consuming, meaning that the availability of water quality data is often insufficient, especially in low-income countries [2]. Citizen science can be a cost-effective way of gathering data, especially with high spatial and temporal resolution, since the long-term costs of acquiring suitable data 'professionally' normally exceed the cost of supporting volunteers to acquire these data [3].

A citizen scientist is a public participant contributing their time and effort towards a scientific study, often in collaboration with or under the direction of professional scientists [4]. Citizen scientists can contribute to water research, increase their scientific understanding of the status of freshwater resources, and more generally learn about environmental issues. Citizen science also changes the way in which the government and institutions interact with the public [5]. A network of dedicated citizen scientists can potentially monitor an entire watershed and generate multi-year water quality data sets [6]. This type of grassroots science has great potential for using local knowledge to identify sources of water contamination, which could ultimately be used to reduce a community's impact on the water body [7].

Water quality monitoring by citizen scientists is particularly useful in rural areas to understand and prevent environmental pollution in water sources used for irrigation [8–10]. However, accurate, affordable, safe, and easy-to-use water testing tools are necessary to engage citizens in water quality monitoring. In a previous study [11], the authors equipped their citizen scientists with colorimetric kits to measure temperature, pH, turbidity, dissolved oxygen, ammonium, and phosphate. Colorimetric kits are easy to use, disposable, and inexpensive [12]. Teichert [12] also provided hand nets and field cards to citizen scientists to collect and identify benthic macroinvertebrates. Citizen scientists can also be trained to visually inspect and record a set of water quality indicators, such as macroinvertebrates including insects in their larval or nymph form, crayfish, clams, snails, and worms [13]. Baalbaki et al. [6] demonstrated the use of a dual-beam spectrophotometer by citizen scientists to measure concentrations of nitrate, iron, and sulfide.

Despite the increasing number of citizen science projects, professionals remain worried about the accuracy of citizen scientist data [11]. According to certain studies, volunteer data differ from data collected by professionals, while in other studies volunteers do just as well as professionals [14]. Such uncertainty can become a barrier to the expansion of citizen science programs and the use of the data from such programs to address water quality issues. Although 73% of the abstracts of studies comparing citizen science data to professional data indicated that the quality of the citizen science data was good, the findings of their quantitative assessment still raised questions about the veracity of the data [15]. In 62% of the studies that provided statistical evaluations with *p*-values, there was no discernible difference between professional and citizen science [6]. In addition, researchers discovered a moderate-to-strong correlation in 51% of the reports, and at least 80% agreement with professional data in 55% of the comparisons reporting percent agreement [11]. Such degrees of accuracy might not be adequate depending on the needs of the researchers. As a result, some researchers are investigating the accuracy of citizen science data and factors affecting the data accuracy to build confidence in data collected by volunteers [11]. For instance, the coefficient of determination can be used to compare volunteer measurements against laboratory measurements. Jollymore et al. [7] compared citizen scientists' and professionals' data using independent (unpaired) t-tests that were applied to the whole data set and found that the accuracy of the data collected by volunteers was shaped by their motivation, knowledge, and expectation.

In Ethiopia, one of the aims of the Water Resources Management Policy is protecting, conserving, and using water resources in a sustainable way [16]. The policy recognizes the need to promote participation of the community in all relevant aspects of water resource management. Unfortunately, many companies in Ethiopia focus on short-term profit at the expense of the river resources with long-term environmental costs for the river catchment [7]. Thus, inclusive action is necessary to prevent the degradation of rivers and to ensure the sustainable management of land and water resources in Ethiopian watersheds. However, regular monitoring of water quality is missing in the country resulting in lack of evidence-based decision making. Large-scale surface water quality monitoring is frequently time-consuming, expensive, and unsustainable, especially for developing nations. Thus, a cost-effective method of water quality monitoring is required,

as indicated by [17], with the application of a 3D-printed IoT-based water quality monitoring system. In this context, citizen scientist participation in water quality monitoring will broaden knowledge and awareness of the value of water resources in the watershed and help implement successful and integrated water quality management plans [18]. To achieve this, the engagement of citizen scientists is of paramount importance because they know much about the pollution sources through their day-to-day interaction with the river. Therefore, the objective of this study is to evaluate the accuracy of citizen science data and use the data to characterize the water quality of the Meki River.

Indicator 6.3.2 of the United Nations Sustainable Development Goal 6 is to increase the proportion of bodies of water with good ambient water quality, and achieving this goal requires monitoring. In Ethiopia, there is a need to involve citizen scientists in water quality monitoring but there is a lack of empirical evidence on how citizen scientists address this urgent monitoring needs, accuracy of citizen science data, and strengths and weaknesses of citizen science approaches with cost-effective, quick, and easy-to-use methods. Therefore, the purpose of this research was to demonstrate how citizen scientists can make a valuable contribution to water quality monitoring and collect reliable water quality data using a low-cost monitoring system to detect pollution sources and assess the suitability of rural water bodies for irrigation purposes. We hypothesized that citizen scientists can collect reliable data with safe and affordable water testing methods.

## 2. Materials and Methods

### 2.1. Description of the Study Area

The Meki River originates from the highlands of the Gurage Mountain and drains into the low-laying Ziway Lake of the central rift valley sub-basin in Ethiopia. The total area of the Meki River catchment is about 2318.58 km$^2$. It feeds 269 M m$^3$ of water to Lake Ziway every year [19]. Geographically, the study area is bounded between 7°51′ N to 8°27′ N and 38°15′ E to 39°02′ E with an elevation of 1636 m above sea level. The climate of the study area consists of three ecological zones: humid to dry humid, dry sub-humid, and semi-arid or arid lands [20]. Temperature and rainfall in the area show strong variations with altitude. The mean annual temperature ranges from about 15 °C in the highlands and around 20 °C in the lowlands. The average annual rainfall varies from around 650 mm in the rift floor to more than 1200 mm in the highlands. The proportions of precipitation that falls during the major rainy season (June to September), the small rainy season (March and May), and the dry season (December to February) are 59%, 28%, and 13%, respectively [20]. In most parts of a year, the rainfall is insufficient to meet the evaporative demand of the catchment. Figure 1 shows the study location, with location details provided in Table 1.

**Table 1.** Location of sampling sites and justifications for their selection.

| Sampling Point & Longitude/Latitude | Justification |
|---|---|
| Site 1 (S1) (38.72°, 8.21°) | This site is in the upstream part of Meki River, which represents natural conditions where agricultural and industrial activities are minimal. Hence, it serves as the reference station that helps to evaluate the background condition of the River water quality. At the site, the river water is used for drinking, cooking, livestock watering, cloth washing, and bathing. |
| Site 2 (S2) (38.83°, 8.15°) | The site is at the downstream part of the Meki River besides the main bridge of the highway from Addis Ababa to Shashemene. Since it is closer to Meki town, the site is polluted by different anthropogenic activities: car washings including trucks, open defecation, wastewater discharges from the town, bathing, and clothing washing site. |

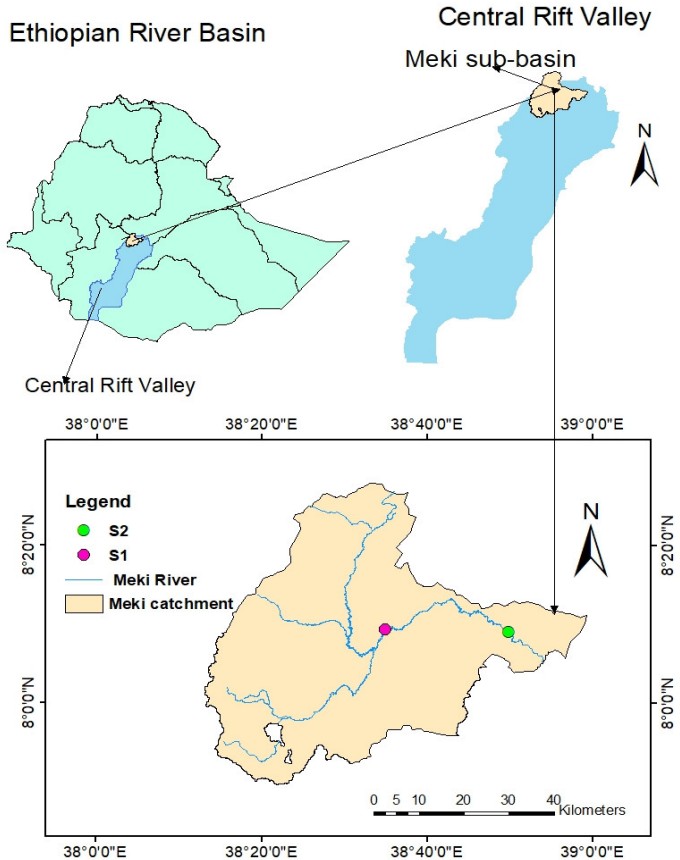

**Figure 1.** Location of the study area and the sampling sites.

Vegetation and Land Uses

The Meki catchment is predominantly cultivated. According to the land use/land cover map from 2018, farmland makes up 81.57% of the catchment area, covering an area of 2858.7 km² (Figure 2). The remaining areas are covered by forest, grassland, water body, shrubland, marshland, and woodland (Table 2). During the dry season, irrigated agriculture provides cash crops, primarily tomatoes and onions. Rainfed farming prioritizes cereals (teff, wheat, maize, barley, sorghum) and false bananas (enset). In the recent decade, widespread and farmer-led irrigation has been observed in the catchment. However, there is weak regulation of chemical inputs to farms and domestic wastewater disposal in the catchment. The Meki River serves as the primary water source for the people living in the Meki catchment. The waste management practice in the catchment area is very poor, in both rural and urban areas [21].

**Table 2.** Types of LULC and area coverage in the Meki catchment.

| Land Use/Land Cover | Area (km²) | % Coverage |
|---|---|---|
| Farmland | 2437.90 | 85.28 |
| Shrubland | 88.55 | 3.09 |
| Grassland | 17.23 | 0.60 |
| Woodland | 3.35 | 0.11 |
| Marshland | 112.44 | 3.933 |
| Forest | 198.07 | 6.92 |
| Water body | 1.06 | 0.037 |

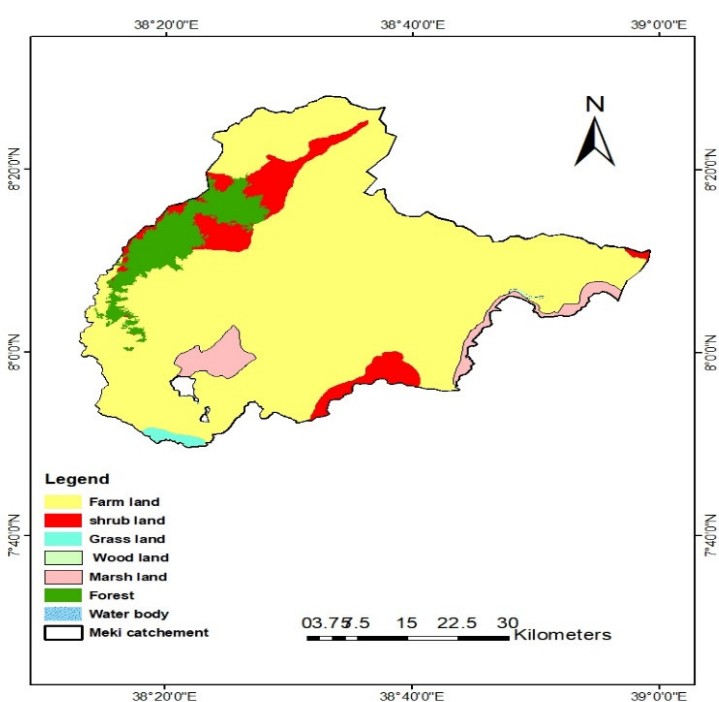

**Figure 2.** LULC map of Meki River catchment.

### 2.2. Sampling Schedule

Sample site selection was based on contrasting pollution sources including intensive agricultural activity, settlements, and waste disposal activities. The Meki River was segmented into an upstream and downstream site. Refer to Table 1 for a description of these sampling sites.

The experts collected the samples from the Meki River from December 2019 to September 2020. pH, conductivity, and temperature were measured onsite and for the rest of the parameters; phosphate, ammonia, nitrate, nitrite, total alkalinity, total hardness, and turbidity water samples were transported with an icebox to Arba Minch University Water Quality Laboratory for analysis.

A 1 L polyethylene bottle was used to collect water for physico-chemical analysis. Before taking the sample, large non-homogeneous matter, such as leaves, rags, twigs, and other floating materials, was removed from the water sample. The bottles were washed with deionized water 24 h before sample collection and rinsed three times with sample water during sample collection. Then the bottles were immersed about 20 cm below the water surface and filled up to the top. Once collected, the samples were tightly closed, labeled, and preserved at 4 °C in an icebox (Mobicool v30 AC/DC, Emsdetten, Germany) and transported to Arba Minch University Water Quality Laboratory. Analysis was performed according to APHA (2005).

### 2.3. Citizen Scientists Selection

To identify and select citizen scientists involved in the study, community stakeholders were consulted, and two citizen scientists (one male and one female) were selected from the upstream (site 1) and downstream (site 2) based on these criteria to select citizen scientists: No previous experience was needed to take part in the study; the criteria included educational background (completed high school and jobless), interest and willingness to participate, commitment to take responsibility, and their residence should be close to the sampling site.

The selected citizen scientists were trained onsite in water quality pollution, benefits of preventing pollution, monitoring water quality, how to test water quality, and the use

of reagents and apparatuses for the analysis at the site (Figure 3). They were also trained in the use of the data recording sheet. After the training, they were regularly supervised by an expert to ensure appropriate sampling and testing. The supervision was conducted through face-to-face meeting and by phone. After training, they collect the samples from respective sampling sites (Figure 4), and then conducted the field measurement using color test strips to indicate the level of water quality for monitoring.

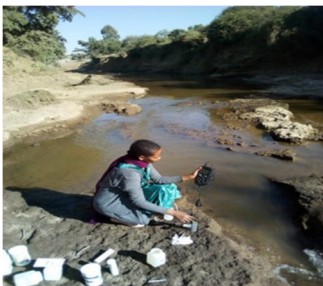
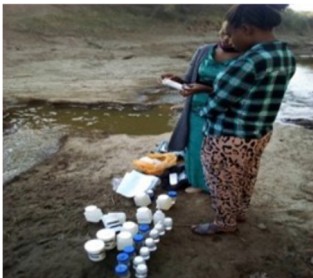

**Figure 3.** Upstream sampling of the Meki River, non-citizen and citizen scientist analyzing water.

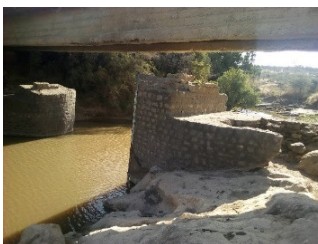
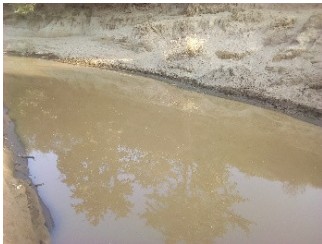

**Figure 4.** Downstream sampling of the Meki River.

### 2.4. Water Quality Analysis by Citizen Scientists

Each of the citizen scientists collected 9 sets of water quality data at S1 and S2, namely: water temperature (°C), ammonia (mg/L), phosphate (mg/L), nitrate (mg/L), nitrite (mg/L), total alkalinity (mg/L), total hardness (mg/L), pH, and turbidity (NTU) using LaMotte test methods. These methods are easy, convenient, cheap, safe, and fast, and can be used by the citizens. LaMotte color test strips (LaMotte, Warwick, UK) are a great way to monitor water without having to use reagents [22]. Ammonia, phosphate, nitrate, and nitrite, total alkalinity, total hardness, and pH were measured by immersing the LaMotte color test strips into the water samples following the manufacturer's instructions. After waiting for 15 s, the color test strips were removed from the water sample and compared with the printed color chart on the test strip box. Turbidity was measured by using a white jar with a Secchi disk by filling the jar with the water sample to the turbidity line on the label. The turbidity chart is held on the top edge of the jar. Looking down into the jar, the citizen scientist compared the appearance of the Secchi disk icon in the jar to the chart. The citizen scientists recorded the result as turbidity in NTU. Water temperature was measured by inserting a mercury thermometer into water and waiting for a few minutes to attain a stable reading, and then the reading was recorded in °C. Figure 5 shows the LaMotte testing system.

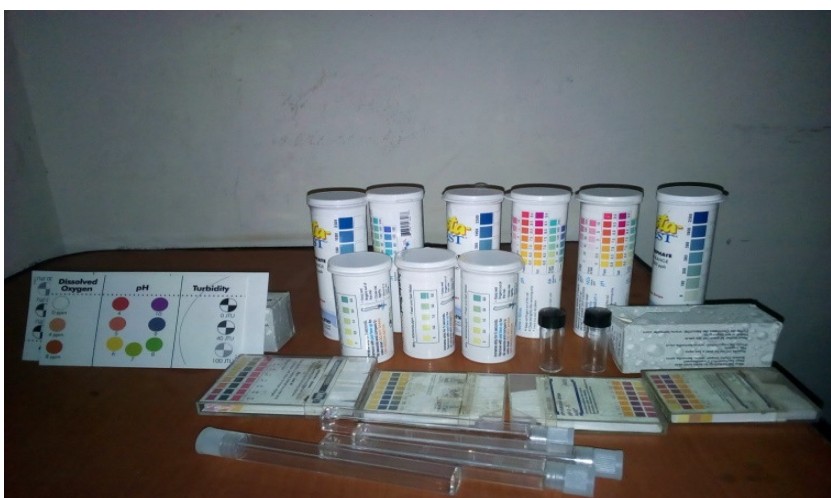

**Figure 5.** LaMotte© tests used to measure field water quality parameter by citizen scientists.

### 2.5. Analysis Using Conventional Method

Electrical conductivity (EC), pH, total dissolved solids (TDS), and temperature (T) were measured in the field using a calibrated portable HACH HQ40D multimeter. The major cation and anion concentrations were measured in the laboratory according to standard methods (APHA, 2005) as summarized in Table 3.

**Table 3.** Methods and instruments used for the analysis of water quality parameters in the lab.

| Variable | Method | Instrument |
| --- | --- | --- |
| Total Hardness | EDTA Titration | Titration set up |
| Total Alkalinity | Titration | Titration set up |
| Turbidity | Nephelometric | Turbidimeter, 2100A, India |
| $NO_3^-$ | Sodium salicylate | UV−VIS Spectrophotometer 2008, India |
| $PO_4^{3-}$ | Stannous chloride | UV−VIS Spectrophotometer 2008, India |
| Ammonia | Distillation | Distillation setup |

### 2.6. Citizen Scientist Data Validation

The accuracy of the citizen scientist data was checked for pH with buffer solutions of pH = 4.0, 7.0, and 9.0. Standard solutions were prepared for each parameter from salts: for phosphate from anhydrous potassium phosphate, for nitrate from sodium nitrate, and for ammonia from ammonium chloride. They were prepared by a laboratory scientist. Also, the conventional methods and citizen scientists' data were plotted against each other to estimate the coefficient of determination ($R^2$ value). The LaMotte color strips' readout by the citizen scientist was plotted on the vertical y axis and the prepared concentrations were plotted on the horizontal x axis.

The relationship was established as $y = mx + b$ for the obtained calibration curve, where '$m$' is the slope and '$b$' is the $y$ intercept. A perfect line would have an $R^2$ value of 1 with $b$=0, and most $R^2$ values for calibration curves are over 0.95 [23].

### 2.7. Statistical Analysis

Analysis was done using MS excel and IBM SPSS-19 version. Analysis of variance (ANOVA) at a 5% level of significance was used to compare the quality of water among all sites and to compare citizen scientists' data with conventional methods during the dry and rainy season. When the $p$-value from the ANOVA and Post Hoc Tukey's tests is below the significance level, which means the $p$-value ≤ 0.05, then the difference is statistically significant, whereas the $p$-value ≥ 0.05 indicates no evidence to support the differences between the two groups. The study design is outlined in Figure 6.

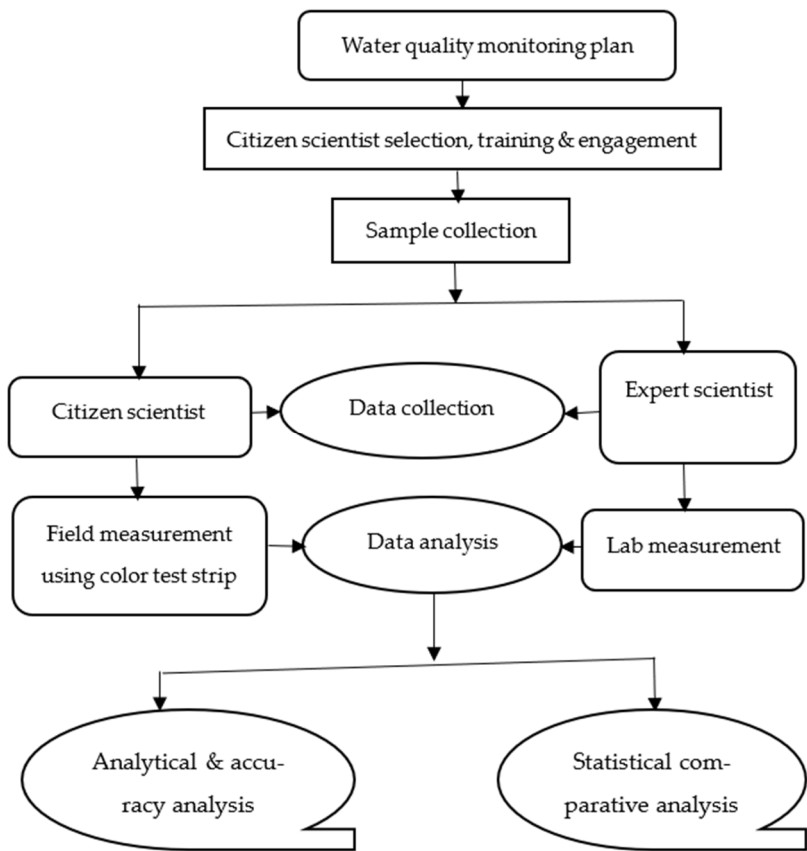

**Figure 6.** Conceptual framework for water quality monitoring tree.

## 3. Results and Discussion

### 3.1. Quality Assurance with Known Standard Concentrations to Assess the Accuracy of Citizen Science Data

The mean values and standard deviations at the two sampling sites for all nine physico-chemical water quality parameters as recorded by the citizen scientists are shown in Table 4. The recorded levels of nutrients ammonia, nitrate, nitrite, and phosphate were all higher in the downstream (S2) as compared to the upstream (S1) sampling location.

**Table 4.** The characteristics of the water quality data collected by citizen scientists.

| Parameter | Upstream(S1) $\overline{X}\pm$ SD | Downstream (S2) $\overline{X}\pm$ SD |
|---|---|---|
| pH | 8.06 ± 0.55 | 7.9 ± 0.57 |
| Temperature (°C) | 20.4 ± 1.5 | 20.71 ± 2.31 |
| Turbidity (NTU) | 40.91 ± 21.53 | 56.82 ± 23.12 |
| Ammonia* (mg/L) | 0.18 ± 0.15 | 0.54 ± 0.41 |
| Nitrate (mg/L) | 11.59 ± 4.19 | 19.32 ± 5.63 |
| Phosphate (mg/L) | 0.2 ± 0.26 | 0.58 ± 0.63 |
| Nitrite (mg/L) | 0.34 ± 0.24 | 0.81 ± 0.67 |
| Total Hardness (mg/L) | 83.18 ± 28.81 | 137.5 ± 45.09 |
| Total Alkalinity (mg/L) | 206.36 ± 51.04 | 168.18 ± 57.37 |

Note(s): * Ionized ($NH_4^+$) and unionized ($NH_3$) ammonia.

The citizen scientist readings using LaMotte color test strips were plotted on the *y*-axis and compared with the values of the known prepared standard solutions on the *x*-axis so that we could compare the results obtained using the LaMotte color test strips

method by citizen scientists with the known concentration of the prepared standard solutions (Figure 7).

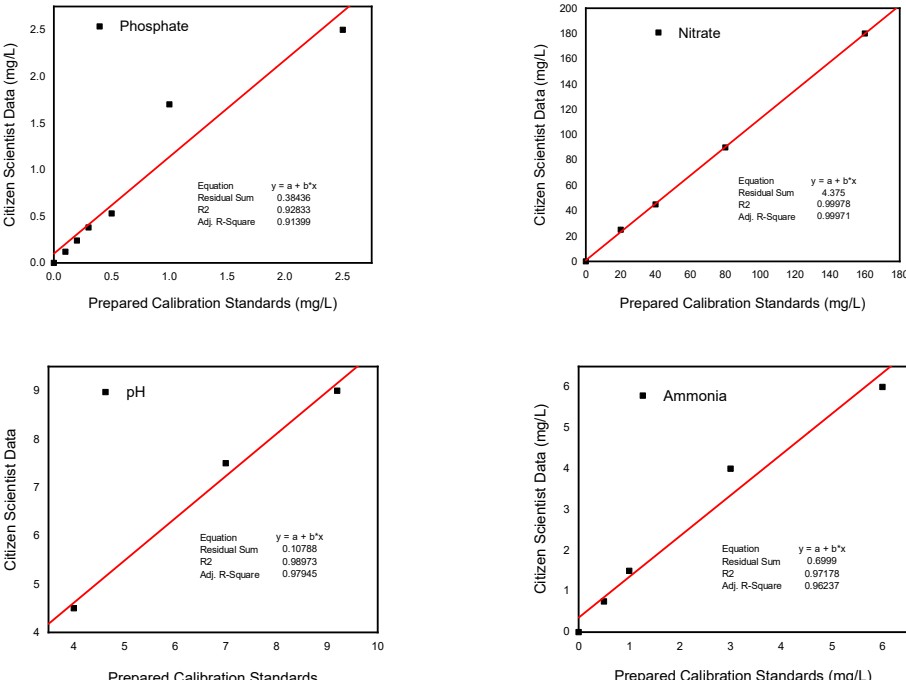

**Figure 7.** Correlation between citizen scientist data and known concentrations for nitrate, ammonia, phosphate, and pH.

The results show that the field water quality data measurements by citizen scientists were accurate as validated by the standard solutions. The $R^2$ values between 0.93 and 1.00 show strong linear relationships, which indicated the two data sets are highly correlated. This demonstrates that citizen scientists can collect accurate data.

The results of citizen scientists' sample readings were also compared with known standard concentrations using % agreement to assess their accuracy. The agreement of the measured (citizen scientist data) as percent of the synthetic standard solution values for phosphate, nitrate, ammonia, and pH were as follows: 123.8 ± 24.7%, 115.6 ± 6.3%, 105.8 ± 7.4% and 133.3 ± 23.6%, respectively. So, the measurements compared reasonably well with the known concentrations, with a tendency to slightly overestimate the true values.

### 3.2. Citizen Scientist Results Compared with Laboratory Measurements by Professionals

In this study, the results (Table 5) compare data generated by citizen scientists in the field with those obtained by conventional laboratory methods. The overall mean for the water quality results, as measured weekly in the field by citizen scientists, and monthly by conventional methods in the laboratory, were compared for the entire data set to reveal any bias (i.e., systematic under or overestimation) between the two approaches.

**Table 5.** Comparison of water quality data measured by citizen scientists using test strips and by professionals using laboratory methods. Values are presented as $\overline{X} \pm$ SD, n.a. means not available.

| Parameter | Season | Citizen Scientists | Conventional Method | $p$ Value (Significance) | (FAO, 1985) * |
|---|---|---|---|---|---|
| pH | Dry | 8.07 ± 0.25 | 8.23 ± 0.41 | 0.81 | 6–8.5 |
| | Wet | 7.07 ± 0.45 | 7.87 ± 0.76 | 0.79 | |
| Temperature | Dry | 21.23 ±1.37 | 23.50 ± 1.37 | 0.08 | n.a. |

|  |  |  |  |  |  |
|---|---|---|---|---|---|
|  | Wet | 18.43 ± 0.18 | 19.50 ± 1.41 | 0.2 |  |
| Turbidity (NTU) | Dry | 64.17 ± 12.84 | 38.33 ± 24.04 | 0.04 | n.a. |
|  | Wet | 94.98 ± 4.70 | 60.00 ± 3.54 | 0.02 |  |
| Total Alkalinity (mg/L) | Dry | 240.00 ± 0.00 | 315.00 ± 65.76 | 0.04 | 0–100 |
|  | Wet | 223.58 ± 5.06 | 127.00 ± 2.83 | 0 |  |
| Total Hardness (mg/L) | Dry | 145.00 ± 21.73 | 128.00 ± 13.73 | 0.47 | 0–500 |
|  | Wet | 67.90 ± 1.77 | 61.25 ± 1.77 | 0.49 |  |
| Ammonia ** (mg/L) | Dry | 1.38 ± 0.76 | 9.78 ± 7.70 | 0.01 | 0–5 |
|  | Wet | 0.13 ± 0.00 | 2.00 ± 0.21 | 0.02 |  |
| Phosphate (mg/L) | Dry | 1.70 ± 0.43 | 1.28 ± 0.67 | 0.33 | 0–2 |
|  | Wet | 1.06 ± 0.43 | 1.66 ± 0.08 | 0.10 |  |
| Nitrate (mg/L) | Dry | 10.50 ± 4.18 | 8.63 ± 1.31 | 0.47 | 0–10 |
|  | Wet | 13.15 ± 0.007 | 12.5 ± 3.5 | 0.99 |  |
| Nitrite (mg/L) | Dry | 1.38 ± 2.14 | 0.57 ± 0.59 | 0.72 | 0–2 |
|  | Wet | 0.11 ± 0.43 | 0.54 ± 0.629 | 0.561 |  |

Note(s): * FAO (Food and Agriculture Organization of the United Nations). ** Ionized ($NH_4^+$) and unionized ($NH_3$) ammonia

In Table 5, if the *p*-value is more than 0.05, it shows no significant difference between the water quality data by the citizen scientist and conventional method. If the *p*-value is less than 0.05, then it indicates a significant difference between the two data sets.

According to the ANOVA results, there was no significant difference between the citizen scientists and conventional method data in both seasons for six of the nine water quality parameters: pH, temperature, phosphate, nitrite, nitrate, and total hardness. However, there were significant differences between the citizen scientists' data and the conventional method for turbidity, ammonia, and total alkalinity, in both dry and rainy seasons. Systematic differences may be due to the timing of sample analysis. During sample storage and transport, fine suspended particles can aggregate and settle out, which could explain the reduced values for the turbidity readings taken in the laboratory. For ammonia, biological transformation of organically bound nitrogen into ammonia via ammonification could explain higher ammonia values recorded by the laboratory method. In addition, the measurement range of test strips used by citizen scientists is limited to 6 mg/L, which may have resulted in underreporting of high ammonia concentrations. For total alkalinity, the average value recorded with the laboratory method in the dry season is also above the maximum range of the color strip method (240 mg/L), which may explain the discrepancy for this parameter. However, in the dry season, the alkalinity values recorded by the citizen scientists are higher than those recorded in the laboratory. Limitations of the test strip methods thus include that citizen scientists could not read values below or above the given measurement range of the test strip color scale, and for phosphate, the color change in the tubes was difficult to assess when the sample was turbid. High concentrations above the measurement can be addressed by sample dilution with distilled water to calculate the result via a dilution factor; however, this requires availability of distilled water and experience, which is challenging for citizen scientists.

*3.3. Water Quality According to Citizen Scientists and Professional Analysis*

3.3.1. pH, Temperature, and Alkalinity

The results obtained from both field and laboratory measurements by a citizen scientist and laboratory expert were compared to FAO 1985 guidelines for irrigation (Table 5).

During both seasons, the recorded value of pH by the citizen scientists and laboratory expert varied slightly between sampling points in the current analysis. In this study, the average pH value determined by a citizen scientist in the field and professionals in the lab was 8.07 and 8.23 in the dry season and 7.07 and 7.87 in the wet season, respectively. The

pH levels during the dry season were greater than those during the wet season. The difference could be due to the pollution caused by degradation of organic matter in the town's wastewater, including from car washes and garages, that flows to the river in the wet season [24]. Additionally, the photosynthetic algae's activities, which absorb dissolved carbon dioxide, might cause the higher average pH values during the dry season [25]. Moreover, low water levels and the presence of fertilizers in the water may contribute to the higher mean value during the dry season. The pH value results in this study agree with the previous finding in the study area [24]. However, for both the dry and wet seasons, the pH values in all the sample points were within an appropriate range according to irrigation guidelines.

The average temperature in this study area measured in the field by both citizen scientists and experts was 21.2 °C and 23.5 °C during the dry season and 18.4 °C and 19.5 °C during the wet season, respectively. These measurements indicate that the river water has an acceptable temperature for irrigation. In general, some variables, including the weather, sample site, and time affect temperature changes, which in turn affect other metrics like the percentage of dissolved oxygen and biological activity [26].

The average value of total alkalinity in this study measured by citizen scientists in the field and experts in the laboratory were 240 mg/L and 315 mg/L in the dry season, and 223.5 mg/L and 127 mg/L in the rainy season, respectively. The alkalinity in both seasons exceeded FAO standards meaning that the river water was not suitable for irrigation purposes. When compared to the FAO standards, the alkalinity value for the Meki River was high. Alkalinity originates mainly from the dissolution of carbonates and will depend on site geology, such as limestone in the catchment. The low alkalinity of rainwater can explain the reduced alkalinity values measured in the wet season.

### 3.3.2. Phosphate, Nitrate, Ammonia, and Total Hardness

The acceptable range of phosphate, ammonia, nitrate, and total hardness for irrigation water varies between 0–2, 0–5, 0–10, and 0–500 mg/L respectively, according to FAO (1985) classification. The average value of phosphate measured by citizen scientists in the field and experts in the laboratory for both seasons were 1.70 mg/L and 1.28 mg/L during the dry, and 1.06 mg/L and 1.66 mg/L during the wet season, respectively. The relatively high mean phosphate value during the dry period was due to the use of soap and detergents by local communities to wash clothes and for bathing in the river, as well as washing cars when the river flow is low, resulting in high values [24]. On the other hand, the low average value measured during the wet season was probably due to the dilution of pollution by the high river discharge. The outcome showed that phosphate concentrations were similar to findings from other water quality studies [21]. Based on their phosphate content, the water sample locations were suitable for use in irrigation according to FAO guidelines.

The average value of nitrate in the study area measured in the field by citizen scientists and laboratory experts were 10.50 mg/L and 8.63 mg/L during the dry season, and 13.15 mg/L and 12.50 mg/L during the wet season, respectively. The higher average values recorded in the wet season could be a result of runoff that carried residential sewage from both rural and urban areas as well as nitrogen-containing fertilizers from neighboring farmland [27]. Based on FAO guidelines, all sample points were suitable for irrigation use in both seasons.

Citizen scientists and experts determined that the average value of ammonia in this research location was 1.38 and 9.78 mg/L in the dry season, and 0.13 and 2.00 mg/L in the wet season, respectively. Ammonia is nutrition for soil bacteria and plant roots when it is added to the soil, and their growth further enriches the soil with nutrients. This promotes the formation of roots with a rich, green hue [25]. However, all sample points were safe during both the rainy and dry seasons according to FAO irrigation guidelines.

The average value of total hardness in the study area measured in the field by citizen scientists and laboratory by experts were 145 mg/L and 128 mg/L in dry season, and 67.9

mg/L and 61 mg/L in wet season, respectively. The highest total hardness value seen in the dry season could be due to the high temperature that enhances the solubility of both calcium and magnesium ions in the water [28]. The lowest average TH value observed during wet season was possibly due to the flow rate of the stream. The dissolved metals were transported to the downstream side when the rivers' flow rate rose, which caused the total hardness content to drop [29]. The average values recorded during both sample periods, however, as determined by both citizen scientists and professionals, were within the permitted limit for irrigation purposes based on FAO 1985 guidelines.

### 3.4. Comparison of the Measured Parameters at Different Sites to Detect Pollution

The results in Table 6 show, the analysis of the water quality parameters and the significant difference between the sites to see if there is pollution in between these locations. These sites were selected as the representative site for the upstream and downstream water quality in the Meki River.

**Table 6.** Physico-chemical water quality analysis results for rainy and dry seasons at sites S1 and S2 as measured in the Meki River catchment by an expert with ANOVA results.

| Parameter | | Sites | | $p$-Values |
|---|---|---|---|---|
| | Seasons | S1 | S2 | S1-S2 |
| Turbidity (NTU) | Dry | 21.55 ± 27 | 53.5 ± 44.5 | 0.00 |
| | Wet | 40 ± 3.53 | 75 ± 3.53 | 0.00 |
| pH | Dry | 8.3 ± 0.4 | 8.1 ± 0.49 | 0.97 |
| | Wet | 7.12 ± 0.12 | 7.65 ± 0.07 | 0.26 |
| $NO_3^-$ (mg/L) | Dry | 7.62 ± 1.9 | 9.47 ± 0.74 | 0.61 |
| | Wet | 12.5 ± 0.71 | 12.5 ± 3.53 | 1.00 |
| $NO_2^-$ (mg/L) | Dry | 0.63 ± 0.88 | 0.5 ± 0.68 | 1.00 |
| | Wet | 0.05 ± 0.2 | 0.54 ± 0.62 | 0.49 |
| $PO_4^{3-}$ (mg/L) | Dry | 1.035 ± 0.9 | 1.49 ± 0.72 | 0.87 |
| | Wet | 1.46 ± 0.09 | 1.85 ± 0.07 | 0.35 |
| $NH_3$ (mg/L) | Dry | 2.75 ± 2.05 | 16.8 ± 17.8 | 0.04 |
| | Wet | 0.75 ± 0.07 | 3.25 ± 0.35 | 0.00 |
| Total alkalinity (mg/L) | Dry | 320 ± 70.7 | 310 ± 98.9 | 1.00 |
| | Wet | 132 ± 2.82 | 122 ± 2.82 | 0.07 |
| Total hardness (mg/L) | Dry | 127 ± 4.24 | 129 ± 29.6 | 1.00 |
| | Wet | 62.5 ± 3.53 | 59 ± 1.41 | 0.59 |

Analysis of variance (ANOVA) at 5% level of significance was used to compare the quality of water among selected sites. The ANOVA is presented in Table 6 and shows a statistically significant difference ($p < 0.05$) for ammonia and turbidity data across the sampling sites. Turbidity at the upstream site (S1) is significantly lower from the site S2 in the downstream. The difference is significant for both dry and rainy seasons. This could be due to the higher value of turbidity observed in the downstream of the Meki River as caused by unprotected overuse of sand extraction from the river, road construction activities, and poor solid waste management. A similar scenario was observed in the Selangor River basin [30]. A higher value of ammonia is observed in the downstream compared to the upstream of the Meki River. This is because there were different activities happening on the river near site S2 including open defecation, wastewater discharge, and body washing, which might contribute to ammonia contamination. Ammonia is a sign of raw excreta and wastewater release [2]. It indicates pollution and the town administration has to promote improved sanitation for the town residents. On the other hand, phosphate, nitrite, nitrate, pH, total alkalinity, and total hardness were not significantly different between the two sampling locations ($p > 0.05$, Table 6) during both the dry and wet seasons.

*3.5. Theoretical and Practical Implications*

Water quality monitoring by using citizen scientists is important in the effort to monitor and maintain water quality data collection. Theoretically, the citizen scientists know more about their surrounding environment, i.e., they know the pollution source and day to day water quality status, and that expands opportunities for scientific study by using endogenous knowledge, such as visual observations of environmental impacts and changes in environments [31]. Moreover, there are many important facts about the water quality of Meki River that impact the ecosystem including human health; this is due to the topography of the area which is exposed to different natural factors and human activities, such as sand extraction, excessive water abstraction, discharge of untreated wastewater, open defection, car washing, and solid waste dumping around the river from the town in the downstream area, due to lack of enforcement and awareness. Return water from extensive and intensive irrigation fields is also further polluting the river water.

The practical implication showed that citizen scientists can investigate the pollution status of the Meki River to create water quality awareness and to protect the river using citizen science approaches, which is very important. Citizen science is making environmental protection more socially relevant while accelerating and enabling participation and open collaboration between communities and researchers; thus, engaging citizen scientists in water quality data collection and cross-checking the accuracy of data is valuable.

## 4. Conclusions

In this work, a citizen science approach was undertaken to check water quality data accuracy of field measurements by comparing data from citizen scientists using LaMotte color test strips with standard solutions for different parameters. The results showed that the field water quality data measurements by citizen scientists were comparable with the standard solution. Nine physical and chemical water quality indicators were tested. The water quality monitoring approach by citizen scientists raised new hope to control the pollution level in the study area. The measurement accuracy of most parameters by citizen scientists and the simplicity of the process of application are making it important for the community. More specifically, there were no statistically significant differences between the results of the citizen scientists and the conventional method (expert) in six out of the nine water quality parameters. This suggests that the accuracy of the citizen science data can be trusted for most parameters. Trained citizens demonstrated that they are equally capable of giving precise, consistent, and reliable physical and chemical water quality test results as academic researchers. However, discrepancies in three parameters were attributed to the timing of sample analysis and the intricacy of dilution procedures required when the results exceed the range of color test strips. The study also revealed that, except for alkalinity, the water quality of the Meki River is within acceptable limits for irrigation at both locations in both the dry and wet sampling seasons.

If the citizen scientist approach is properly implemented, well trained citizen scientists can collect water quality data to identify the source of pollution and monitor the status of water bodies, especially where resources are limited and in remote areas.

**Author Contributions:** Conceptualization, W.Z.B., A.T.H. and K.K.A.; methodology, W.Z.B., A.T.H., K.K.A. and D.D.K. software, W.Z.B. and D.D.K.; validation, A.T.H., K.K.A., D.W. and K.A.; formal analysis, W.Z.B. and D.D.K.; investigation, W.Z.B., A.T.H., K.K.A. and D.D.K.; resources, A.T.H., D.W. and K.A.; data curation, W.Z.B., A.T.H., K.K.A., D.D.K., D.W. and K.A.; writing—original draft preparation, W.Z.B. and D.D.K.; writing—review and editing, A.T.H., K.K.A., D.W. and K.A.; visualization, A.T.H., K.K.A., D.W. and K.A.; supervision, A.T.H., K.K.A., D.W. and K.A.; project administration, A.T.H., D.W. and K.A.; funding acquisition, A.T.H., D.W. and K.A. All authors have read and agreed to the published version of the manuscript.

**Funding:** This project was funded, through the Water Security and Sustainable Development Hub fund, by the UK Research and Innovation's Global Challenges Research Fund (GCRF), grant number: ES:/8008179/1, and the Royal Society, grants ICA/R1/191241. We also would like to thank the

school of graduate studies, Arba Minch University for the support of financial and laboratory facilities.

**Data Availability Statement:** Additional data is available upon request.

**Acknowledgments:** Citizen scientists contributed to the data collection for this study.

**Conflicts of Interest:** The authors declare no conflict of interest.

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
