# Peer review of "Citizen Science for Water Quality Monitoring in the Meki River, Ethiopia: Quality Assurance and Comparison with Conventional Methods"

_water, doi:10.3390/w15020238_

Round 1

Reviewer 1 Report

Comments to authors:

The authors included and performed citizen science for addressing inadequate water quality monitoring in the Miki River, Ethiopia. Overall, the manuscript has a concrete structure, with presentable results and discussions. However, there are a few comments to be addressed.

1.     Abstract: Include a line or two about the novelty and the research gap that you are addressing.

2.     Introduction section: The research gap and the research objectives were not clear in the submission. A clear list of previous studies should be provided to clearly identify the research gap in the research and also highlight the novelty of the research. 

3. Unclear about the "small rain" season. Include the precipitation rate.

4. Figure 2: Meki River basin can be further modified and optimize for the size of the map.

5. Sampling S1 and S2 points do not correspond to Site 1 and Site 2. Provide the abbreviation.

6. The concept of citizen science is not understood here. The education for the local citizen was not mentioned here and seems just the two citizens were just asked to perform sampling. 

7. The current framework is not clear about the participation of citizens in this study. More appropriate discussion should be included else it is just a conventional water quality analysis paper. Suggest providing a framework structure for this concept.

8. Discussion section: Not many in-depth comparisons were done. I would suggest the authors compare the findings with existing literature. Unique contributions of research should be highlighted.

9.     Implications: The authors must develop a subsection for theoretical and practical implications. Implications could be enhanced by providing the results of your work toward the development and adoption of the current findings.

10. Some relevant studies which can improve the discussion of the manuscript are missing in the study:

(b) DOI: 10.1016/j.jclepro.2021.129230

(c) doi: 10.1007/s10661-020-08543-4

11.     Conclusion section seems to be a repetition of the results section. Huge modifications are requiredPlease make sure your ‘conclusion’ section underscores the scientific value added to your paper, and/or the applicability of your findings/results, as indicated previously. Please revise your conclusion part into more detail. Basically, you should enhance your contributions, and limitations, underscore the scientific value added to your paper, and/or the applicability of your findings/results and future study in this section.

Author Response

Dear Reviewer, We really appreciate you for giving your precious time and valuable suggestion on the manuscript. We have taken very careful consideration of your questions and comments to improve the manuscript in all aspects. In the revised manuscript, all the corrections and changes are written accordingly and highlighted in yellow color. Detailed point-by-point responses are given 

Author Response

(The authors gave the same response as above.)

Round 2

Reviewer 1 Report

The authors have substantially addressed and improved the manuscript. The current manuscript fits the quality and standard of the manuscript of the journal, therefore, I would recommend "ACCEPTING" the manuscript.